# The Italian Research on the Molecular Characterization of Maize Kernel Development

**DOI:** 10.3390/ijms231911383

**Published:** 2022-09-27

**Authors:** Gabriella Consonni, Giulia Castorina, Serena Varotto

**Affiliations:** 1Dipartimento di Scienze Agrarie e Ambientali (DiSAA), Università degli Studi di Milano, Via Celoria 2, 20133 Milano, Italy; 2Department of Agronomy, Food, Natural Resources, Animals and Environment (DAFNAE), Università degli Studi di Padova, Viale dell’Università 16, 35020 Legnaro, Italy

**Keywords:** maize, kernel, endosperm, seed mutants, seed development, zeins

## Abstract

The study of the genetic control of maize seed development and seed-related pathways has been one of the most important themes approached by the Italian scientific community. Maize has always attracted the interest of the Italian community of agricultural genetics since its beginning, as some of its founders based their research projects on and developed their “schools” by adopting maize as a reference species. Some of them spent periods in the United States, where maize was already becoming a model system, to receive their training. In this manuscript we illustrate the research work carried out in Italy by different groups that studied maize kernels and underline their contributions in elucidating fundamental aspects of caryopsis development through the characterization of maize mutants. Since the 1980s, most of the research projects aimed at the comprehension of the genetic control of seed development and the regulation of storage products’ biosyntheses and accumulation, and have been based on forward genetics approaches. We also document that for some decades, Italian groups, mainly based in Northern Italy, have contributed to improve the knowledge of maize genomics, and were both fundamental for further international studies focused on the correct differentiation and patterning of maize kernel compartments and strongly contributed to recent advances in maize research.

## 1. Introduction

The study of the genetic control of seed development and seed-related pathways has been one of the most important themes approached by the Italian scientific community, and some of the Italian research groups interested in this topic chose maize (*Zea mays* L.) as the main species to carry out their activities. This species has always attracted the interest of the Italian community of agricultural genetics since its beginning, as some of its founders based their research projects on and developed their “schools” by adopting maize as a reference species. Some of them spent periods in the United States, where maize was already becoming a model system, to receive their training [1].

Angelo Bianchi, the leading maize scientist, was the first to travel to Cambridge, Massachusetts, where he spent a period, between 1954 and 1956, in the laboratory of Paul Mangelsdorf at Harvard University. He brought his expertise to Italy, where he founded the “Bianchi school” and taught maize genetic methodology to students at Pavia University and Piacenza University (Figure 1). At the end of the 1950s, Angelo Bianchi had the opportunity to move from the University of Pavia to the University of Milan. At the Institute of Genetics directed by Claudio Barigozzi, a professor of genetics who had a significant role in promoting this discipline, an important project on the study of the effect of ionizing radiations was started, and was the main reason for his transfer. The project was supported by Felice Ippolito of CNEN [2].

Starting from the Bianchi school, other schools dedicated to maize genetics were opened by Bianchi’s students, including Francesco Salamini and Giuseppe Gavazzi, who received their training, respectively, from O. Nelson at Purdue and R. A. Brink in Madison, Wisconsin [1]. The contributions of their research groups are briefly mentioned hereafter, along with those of other Italian teams that focused their research activities on the maize kernel.

Maize genetic analysis has also received the contributions of other Italian teams, whose activities have covered a variety of different themes. Due to space constraints, only some selected examples are reported here below. Cuticle biosynthesis was first characterized through the isolation of glossy mutants [3,4], and more recently with the characterization of a MYB regulatory gene [5]. The roles of R/bHLH and C1/MYB transcription factors, whose activities are required for promoting the transcription of structural genes involved in anthocyanin pathway, were described [6,7], and a locus-controlling flowering time was isolated [7]. Contributions to the analysis of genome architecture and heterosis were also provided by the work of Morgante et al. [8] and Frascaroli et al. [9]. More recent examples include the dissection, through QTL and genomic analyses, of root architecture and leaf development [10,11]. The issue of the security of maize products has also been tackled with the analyses of the susceptibility to ear-root diseases and related mycotoxin contamination [12,13]. LPA (low phytic acid) mutants involved in phytic-acid biosynthesis were detected [14], and the application of these mutants with high levels of free phosphate in the seeds to enhance the nutritional value of crops was analyzed [15].

The identification of the genetic bases of maize traits is supported by the availability of a population of recombinant inbred lines [16], and extensive genetic resources are available at CREA (Centro di Ricerca Cerealicoltura e Colture Industriali) in Bergamo. This research institute, also referred to as the “Maize Experimental Station”, established in 1920, has historically contributed to maize breeding and genetics, and retains a fundamental role in preserving maize biodiversity, since it hosts a germplasm bank that includes numerous inbred lines and local varieties. The same institute hosts MAYDICA (https://journals-crea.4science.it/index.php/maydica/index, accessed on 26 August 2022), a journal founded in 1956 specifically dedicated to the field of the genetics, molecular biology, biochemistry, physiology, breeding, and agronomy of maize and allied species.

The aim of this report is to illustrate the research work carried out in Italy by different groups that studied the maize kernel and underline their contributions in elucidating the fundamental aspects of caryopsis development through the characterization of maize mutants. Initially, we describe maize seed development and some mutants that have been characterized in Italian labs. Then, we focus on the contribution of Italian research on genes and molecular mechanisms controlling maize endosperm development and composition (Table 1).

## 2. Genetic Molecular Analysis of Seed Development in Maize

Maize (*Zea mays*) is one of the most important crops cultivated worldwide as a source of animal feed, human food, and industrial products. It is also considered as a keystone model plant system for basic and applied research. Some maize features that made this species an attractive and valuable model organism have been appreciated for many decades by a wide international scientific community. The maize plant exhibits separated male and female flowers of considerable size, a feature that greatly facilitates the conduction of controlled pollinations, which are essential in genetic analysis. In addition, several hundred seeds can be produced from a single pollinated ear, and each maize kernel is a large, single-seeded fruit suitable for phenotypic analysis. Kernel traits have been instrumental for the discovery and characterization of important biochemical and developmental pathways, as well as genetic molecular mechanisms. It is worth mentioning that Barbara McClintock based her discovery of transposable elements on the observation of mosaic sectors induced by the activity of these elements in the aleurone, the anthocyanin-enriched outer cell layer of the maize endosperm [17]. The relatively large size of the organs and tissues of the maize kernel has greatly facilitated cytological cellular and molecular studies. Discrete cell layers can be easily visualized from fixed tissues immobilized on glass slides and eventually dissected and used for single gene expression and “omic” studies. Genome-wide expression profiles of kernel tissues and of embryo/endosperm interfaces have therefore been obtained [18].

Other maize peculiarities include the large physical size of maize chromosomes that has greatly facilitated cytogenetic research. The presence of chromosomal knobs simplified the tracking of chromosomes during cell divisions; aberrations, such as translocations and inversions, allowed for the ascribing of genes to chromosomal locations [19]. From a genomic point of view, it has been shown that maize has an extraordinary level of diversity. Being a naturally outcrossing species, its genetic architecture is more similar to other outcrossing organisms, such as humans [20]. On the other hand, maize retains, as other plant species, the ability to self-cross and produce F_2_ populations, including homozygous individuals of interest for genetic analysis.

Data, findings, and models produced in maize can be translated to other cereals, including rice (*Oryza sativa*), sorghum (*Sorghum bicolor*), wheat (*Triticum* spp.), and barley (*Hordeum vulgare*). Contrariwise, maize research can benefit from achievements obtained in related cereal species.

As already mentioned, this review is focused on the genetic dissection of seed development and related metabolic pathways. Significant achievements will be illustrated, with particular emphasis on the contribution of Italian research.

### 2.1. Seed Origin and Development

In maize, as with other cereals such as barley, rice, and wheat, seeds are complex biological systems comprising three genetically distinct components nested one inside another: the embryo, which gives rise to the future plant; the endosperm, a nutrient-rich storage tissue; and the surrounding maternal tissues. External tissues are of maternal origin, and include the nutritive nucellus, which degenerates as the endosperm increases its size, and the pericarp, a protective layer which comprises the pedicel at the basal pole [21,22]. The seeds are responsible for initiating the new sporophytic generation of the plant, and are also an important source of nutrients. The comprehension of their biology and genetics, which has stimulated many research studies, can be a prerequisite for biotechnological applications and genetic improvement of the agricultural production. Many attempts have been made to elucidate the genetic basis of kernel size and weight, two important yield components, as well as the nutritional value of their storage products [23].

Maize seed, like those of all other angiosperms, is initiated by the double fertilization process in which two male gametes (sperm cells), which are genetically identical, are released upon fertilization into the female gametophyte, the embryo sac [24,25]. The nucleus of one sperm fuses with the egg cell and produces the diploid embryo, while the nucleus of a second sperm fuses with the two polar nuclei of the embryo sac and gives rise to the triploid endosperm. The embryo and endosperm are genetically identical, except for their ploidy levels, so the ratio between the maternal and the paternal contribution differs in each compartment. The embryo is diploid, and the endosperm is triploid. It is well known that disruption of this balance can lead to seed abortion. In other words, a 2:1 ratio of maternal to paternal genomes is necessary for proper endosperm development and, consequently, for correct seed development [26].

The three distinct kernel compartments, i.e., embryo, endosperm, and maternal tissues, undergo a tightly coordinated development that can be divided into three main phases: early development, filling, and maturation [24,25]. During early maize seed development, embryo and endosperm cells differentiate into populations forming distinct tissues and organs [21,27]. Then, during the filling stage, from ~12 to ~35 days after pollination (DAP), both the scutellum of the embryo and the central region of the endosperm accumulate reserve substances, while the surrounding maternal tissues provide or transport the necessary nutrient supplies. Finally, the kernel enters a maturation phase, lasting from ~35 to ~50 DAP, during which the kernel dehydrates and enters quiescence prior to dispersal [28,29]. It has been shown that the three phases are characterized by distinct genetic programs that determine their progression. In addition, the parallel growth and developmental changes of the three kernel compartments rely on a complex exchange of signals among them [30,31].

### 2.2. Differentiation of Embryo and Endosperm Domains

The mature maize kernel has well-defined architecture with an embryo that appears enclosed within the endosperm, which occupies about 70% of the kernel volume and that is itself surrounded by the pericarp. The maize embryo consists of the embryo proper and the scutellum. The first comprises a well-differentiated embryonic axis containing root and shoot primordia, which are located at opposite poles, and a stem with five or six internodes, bearing a leaf at each node. It is surrounded by the scutellum, a shield-shaped organ considered a single massive cotyledon, in which nutrient reserves accumulate [32]. Analysis of the maize embryo development shows that at 4 DAP, two distinct parts can be distinguished: an apical embryo proper and a cone-shaped basal part, namely the suspensor, which will degenerate at the end of the early development [31,33]. At around 8 DAP, during the transition stage, a shift from a radial to a bilateral symmetry occurs, and the above-mentioned scutellum is produced at the abaxial side of the embryo proper. Subsequently, during the coleoptilar stage (9–12 DAP), the shoot apical meristem develops on the adaxial side, marking the apical pole of the future embryonic axis and the root apical meristem differentiates within the embryo body, defining the basal pole of the embryonic axis. Shoot and root meristems will be surrounded by the protective coleoptile and coleorhiza, respectively [21,28,32]. In particular, the work of Giuliani et al. [33] showed that the occurrence of programmed cell death (PCD) is a key component of embryo development and is confined to organs that do not contribute to the plant body. Between 14 and 20 DAP, during the elaboration of the primary shoot and root axis, the cells of scutellum, coleoptile, root cap, and the suspensor exhibit distinct signs of programmed death i.e., TUNEL-positive nuclei and genomic DNA ladders.

In the early developmental phase (4–6 DAP), endosperm nuclei undergo proliferation, producing a coenocyte that becomes cellularized [27,34] and then differentiates into distinct domains characterized by unique cellular morphologies and gene expression patterns. Domains are specified and occupy specific territories already within the early endosperm development [29,34]. The basal endosperm transfer layer (BETL) and the aleurone layer (AL), located at the endosperm periphery, are in contact with the maternal tissues.

BETL is the zone of the endosperm closest to the phloem terminals in the pedicel, and is made of specialized cells containing numerous cell wall ingrowths that amplify the basal membrane surface [35]. BETL cells allow for the uptake of the nutrients from the mother plant to the endosperm. An interesting feature of differentiated BETL cells is the polarization of cell wall ingrowths that develop in the apical–basal axis. Their development is likely to be induced by incoming sugars and/or other diffusible signals, such as auxins, from the maternal tissues. The molecular and physiological processes underlying the development of this layer have always attracted the attention of researchers, most probably for their essential roles in the grain-filling process. The specific contribution of the Italian laboratories will be presented later.

The AL is an epidermis-like single cell layer surrounding the endosperm and separating the endosperm from the maternal tissues. It is necessary for the mobilization of stored reserves during germination (reviewed in [36]). In the aleurone cells, where anthocyanins are accumulated, regulatory and structural genes are expressed in a cell-specific and time-specific pattern, as elegantly shown in the work of Procissi et al. [37]. By using in situ hybridization, the authors revealed a correlation between the activation of structural genes and anthocyanin accumulation. They also showed that transcriptional control of the structural genes within the pericarp was exerted by *Sn1* (*Scutellar node1*) and *Pl1* (*Purple plant1*) genes, whereas in the aleurone it was promoted by *R1* (*Colored1*) and *C1* (*Colored1 aleurone1*) genes. Both *Sn1* and *R1* encode for bHLH regulatory proteins, while *Pl1,* as well as *C1,* encode for MYB proteins. In addition, the *Sn* expression was shown to be enhanced by light, whereas the *R* gene expression was not.

Two endosperm domains are in contact with the embryo. The embryo-surrounding region (ESR), consisting of densely cytoplasmic cells with small vacuoles, surrounds the embryo at the early stages (7–9 DAP), and later (12 to 15 DAP) persist exclusively in a small region around the suspensor portion of the embryo [25]. The zone of the endosperm adjacent to the scutellum (EAS) is detectable when the scutellum emerges (~9 DAP), and persists throughout embryo growth (up to ~20 DAP) [18].

The central portion, referred to as central starchy endosperm (CSE), is composed of large cells whose development implies different phases characterized by peculiar developmental processes. Starting at around 8–10 DAP, these cells gradually switch from a mitotic cell cycle to endoreduplication. During this phase, reserve substances, i.e., starch and proteins, start to accumulate, and the CSE grows dramatically [38]. Among the regulatory components controlling the progression of this phase there are retinoblastoma-related (ZmRBR1) proteins, whose role is depicted here below. Subsequently, the grain-filling phase initiates and lasts for a long period (~15 to ~45 DAP), followed by a period during which grain-filling ceases and the central region undergoes progressive programmed cell death [38]. Starch is made of two α-glucan polymers, linear amylose and branched amylopectin, that are packed into semi-crystalline granules in amyloplasts. Prolamins, or zeins, the principal endosperm storage proteins, are found in protein bodies. The time-specific and coordinate activation of genes involved in the CSE development progression and biosynthetic pathways implies distinct regulatory networks (revised in [23,39]). As illustrated here below, the contribution provided by the Italian groups led to the identification of the main actors involved in the regulation of storage protein synthesis and deposition and the comprehension of the genetic, as well as epigenetic, regulatory mechanisms involved.

### 2.3. The Mutant Approach: Type of Mutants and Usage in Functional Gene Studies

Since the 1980s, most of the research projects aimed at the comprehension of the genetic control of seed development and the regulation of storage products’ biosyntheses and accumulation have been based on forward genetics approaches. Although quite a variety of mutants have been detected in maize that affect all organs [40] (https://www.maizegdb.org/data_center/phenotype, accessed on 26 August 2022), the collections are particularly rich in variants related to kernel development. Because of the large maize kernel, the characterization of many morphological and biochemical mutants was straightforwardly performed, and their molecular genetic analysis provided insight into the mechanisms of embryo and endosperm formation and metabolic pathways. The forward genetics approach was also adopted by many Italian laboratories, where mutant collections were obtained by means of chemical, as well as transpositional, mutagenesis. A broad classification based on mutant kernel architecture includes three main categories: (i) *defective kernel* (*dek*) mutants impaired in both the endosperm and embryo [41]; (ii) *embryo-specific* (*emb*) mutants characterized by reduced or arrested embryo development but relatively normal endosperm formation [42,43]; (iii) *defective endosperm* (*de*) or *endosperm-specific* mutants with more-or-less normal embryo development, but defective endosperm (Figure 2) [44].

The *emp* (*empty pericarp*) mutants represent a subclass of *dek* mutants with the most severe reduction in endosperm development and tiny embryos (Figure 2D). Homozygous mutant *emp* kernels contain little, if any, endosperm and a tiny embryo, and are easily recognizable in segregating mature ears because they are flattened by compression from the surrounding normal seeds [45,46]. The work of Sangiorgio et al. [47] analyzed a group of *emp* mutants obtained through transpositional mutagenesis and showed that in two cases the mutant phenotype was modified by the genetic background. While exhibiting the classical emp phenotype in other lines, emp*-8376 and emp4-9475 mutations, if transferred to the A636 and Mo17 inbred lines, respectively, both showed a significant increase in endosperm size. Since the two phenotypes showed a Mendelian segregation in F_2_ and F_3_ progenies, it was speculated that the improved phenotype might be the result of an interaction between the *emp* mutant and a *suppressor* that has no obvious phenotype of its own. Significant subcategories of the *de* group include the *sugary* and *opaque* mutants impaired in storage starch and protein biosynthesis, respectively (Figure 2E,F). Molecular characterization of the opaque phenotypes has been effective for gaining insight into the molecular mechanisms controlling protein deposition in the CSE, as illustrated afterwards in this review.

A wide group of spontaneous mutants showing a smaller seed size was also detected in Italian maize populations in the 1960s [44]. The collection, which was enriched with mutagenesis-induced mutants, is still maintained and exploited at “Università di Piacenza”. Mutant collections are also present at “Università degli Studi di Milano” and “Università di Bologna”.

Seed mutants have been classically adopted for linking genes and phenotypes and thus isolating and characterizing the genes involved. In this context, two PPR (pentatricopeptide repeat protein) genes, i.e., *emp4* [46] and *PPR8522* [48], were characterized in collaboration with other European groups. The product of *Emp4*, the first nuclear-encoded PPR protein characterized in maize, is localized in the mitochondria, where it regulates the expression of a small group of transcripts. The mutant study showed that EMP4 is required for maintaining mitochondrial populations and promoting cell wall ingrowths differentiation in the BETL cells. It was thus speculated that the lack of endosperm growth in the mutant was a consequence of the absence of a normal BETL development. PPR8522 is a chloroplast-targeted PPR protein that is necessary for the transcription of nearly all plastid-encoded genes. The loss of PPR8522 confers an embryo-specific (emb) phenotype, which was characterized at the Université Lyon, France, in collaboration with an Italian team. The mutant’s endosperm was normal, while mutant embryos appeared to be reduced in size and exhibited morphological aberrations. The study was the first to associate the loss of a PPR gene and consequent alterations in plastid development with an embryo-lethal phenotype in maize.

Other mutants isolated by Italian researchers, which contributed to deciphering the genetic control of seed development and metabolism, will be discussed in subsequent chapters.

Lately, reverse genetic approaches have been integrated to achieve the same goal, as shown in the work of Forestan et al. [49] mentioned here below, that revealed the importance of histone deacetylases genes in endosperm progression. Recent advances in genomics technologies have provided new meanings to the mutant studies, since genetic and phenotypic analyses can be integrated with genomic data. For instance, transcriptome studies of developing mutant and wild-type seeds have greatly enhanced the capability to attribute genes to networks or signaling cascades and thus obtain a deeper insight into their functions.

## 3. A Focus on Endosperm Development

### 3.1. Endosperm Storage Protein-Accumulation and Transcriptional Regulation

The Italian molecular research on maize endosperm storage protein deposition dates back to the beginning of 1980s, with the pioneering studies performed on a group of alcohol-soluble polypeptides collectively known as zeins and the almost contemporary identification of their main regulatory gene known as *opaque endosperm2* (*o2*) [50,51].

Zein storage proteins, which account for 50% of the total endosperm proteins in the maize seed, are synthesized in the endosperm tissue between 15 and 40 days after pollination (DAP) on the rough endoplasmic reticulum and on the surface of protein bodies of the central endosperm [52]. In 1982 at CNR in Milano, some cDNA clones, representative of the two major classes of zein, of 20 and 22 kd, respectively, were initially characterized [53]. Light-chain zein genes were located on chromosomes 4, 7, and 10, whilst genes coding for some of the heavy-chain zeins to the distal part of the long arm of chromosome 4. Intriguingly, zein genes present as multigene families, are organized in clusters on different chromosomes in the maize genome, and are coordinately and specifically transcribed only in endosperm-central cells. A specific and extensive undermethylation of zein sequences occurs in the endosperm, while a common methylated pattern is detected in the different somatic tissues and in the embryo [54]. DNA methylation is an epigenetic mark that is often associated with gene silencing and imprinting expression. By studying zein gene expression, [55] reported that some zeins genes are maternally imprinted at DNA, RNA, and protein levels, and this maternal specific mechanism of expression seems to be confined to the endosperm. More-recent genetic studies demonstrated that the uniparental expression of α-zeins depends on a specific combination of parental genetic backgrounds, while the DNA methylation at zein loci did not show an obvious correlation with expression, indicating that additional factors determine the parent-of-origin effects at these loci [56,57]. Overall, the genetic analyses revealed that parent-of-origin expression of specific α-zein alleles relied on a complex interaction between genotypes in a similar manner of paramutation-like phenomena [57].

The initial studies on the zein system also clarified the relationship between sequence types and polypeptide size classes and paved the way to the possibility of inserting lysine codons in their sequence to raise the nutritional value of these seed-storage proteins [58].

In 1989, a collaboration between CREA-Bergamo and the Max Planck Institute in Cologne produced the first sequence of both the genomic and cDNA clone of *Opaque 2* [59], one of several loci with regulatory effects on the synthesis of zein proteins during endosperm development. The scientific attractiveness of *Opaque2* (*O2*) locus and phenotype relied on in the observation that in its recessive mutant, the nutritionally unbalanced zein proteins are reduced by up to 60%, and the 22 kd zein class component is almost absent. The zein content reduction improves the nutritional composition of the kernel and makes the endosperm more digestible to human and monogastric animals. Subsequently, the protein sequence of the basic leucine zipper transcriptional-activator encoded by the *O2* was reported to possess an unusually long 5′ leader sequence, containing three upstream open-reading frames which interfere with its transcription in the mutant [60,61]. Numerous recessive *O2* alleles of independent origin were characterized at the highly polymorphic *O2* locus [62,63,64]. The comparison of genomic sequences spanning the first exon and obtained from a series of wild-type and recessive alleles revealed the presence of a hypervariable region in the N-terminal part of the O2 protein. It was also observed that the O2 protein exists in vivo as a pool of differently phosphorylated polypeptides, and O2 DNA-binding activity is modulated by a phosphorylation/dephosphorylation mechanism that appears to be influenced by environmental conditions [65]. Indeed, O2 activity was shown to be downregulated at night by both a reduction in *O2* transcript and by the hyperphosphorylation of residual O2 proteins [66]. This observation led the authors to hypothesize that the regulatory gene activity during endosperm development may be sensitive to a diurnal signal(s) emanating from the plant and passing into the developing seeds [66]. Additionally, it was observed that unlike the majority of plant transcriptional activators, O2-binding activity is not impaired by CpG DNA methylation. However, the pattern and average level of C-methylation could modulate DNA binding affinity of the O2 protein to its targets [56,67].

Together with the *O2* mutations, any of three semi-dominant or dominant mutations *floury2* (*fl2*) [68], *Defective endosperm*-B30 (*De-B30*), and *Mucronate (Mc1),* with an opaque phenotype, reduced the levels of zein, increased the lysine content in the endosperm, and altered the levels of HSP70-like proteins, suggesting a role for these proteins in the transport and/or accumulation of zein into protein bodies [69].

In the developing endosperm, the maize O2 also controls the expression of an abundant albumin, coded by the *ribosome-inactivating protein1* (*rip1*) gene, initially termed b-32 [70,71]. The promoter of the *b-32* gene is activated by *O2* gene product and contains five O2-binding sites (GATGAPyPuTGPu). Two of these sites are embedded in copies of the ‘endosperm box’, a motif thought to be involved in endosperm-specific expression, which is also represented in 22 kd zein promoters. The O2 protein was also shown to bind in vitro and activate in vivo its own promoter [72].

Later works indicated that O2 is involved in the regulation of another endosperm protein, the *pyruvate orthophosphate dikinase-1* (*cyPPDK1*) [60,73]. The regulation by the O2 locus of *cyPPDK1* and control of alpha-zein synthesis by O2 suggests that the O2 protein may play a more general role in maize endosperm development than previously thought [60].

Currently, the regulation of zein storage proteins biosynthesis and protein body formation in kernels is still an important topic in maize research. Zeins lack two essential amino acids, lysine and tryptophan, and current projects aim at overcoming this deficiency that limits the use of maize proteins in the food and feed industries [74].

### 3.2. Chromatin and Epigenetic Regulation of Endosperm Development

A major challenge in studies of endosperm development is the elucidation of essential regulatory steps in the expression of nuclear genes that encode proteins and involve a complex set of nucleic acid–protein and protein–protein interactions. In this context, chromatin structure and remodeling are known to play a pivotal role, and numerous factors capable of altering the structure of chromatin have been identified [75]. A well-recognized important mechanism for the dynamic modification of chromatin structures is the acetylation and deacetylation of histone residues, mediated by the activities of histone acetyltransferase (HAT) and histone deacetylase (HD) [76]. Histone hyperacetylation usually correlates with transcriptionally active genes in open chromatin conformation, whereas hypoacetylation correlates with transcriptionally repressed heterochromatin [75]. Several Italian studies have contributed to the clarification of the correlation between the level of histone acetylation and the transcriptional activity during endosperm development in maize.

ZmHDA101 (initially named ZmRpd3I) was the first histone deacetylase homologous to yeast RPD3 gene to be cloned and analyzed from a plant [77]. This gene is expressed in the endosperm, starting from a few DAP until the completion of grain-filling. Both *ZmRpd3I* and the *retinoblastoma-related homologue (ZmRBR1*) are expressed during endosperm development and were shown to interact in vitro [78]. Furthermore, retinoblastoma-associated protein1 (ZmRbAp1), a maize member of the MSI/RbAp family also expressed in the endosperm [79], was demonstrated to facilitate these proteins’ interaction [78]. These findings represented the first indication that a regulator of important biological processes, such as ZmRBRI, can recruit a histone deacetylase, ZmRpd3I, to control gene transcription in plants. Additional investigations showed that the maize Rpd3-type histone deacetylases genes *ZmRpd3/101*, *ZmRpd3/102*, and *ZmRpd3/108* (later renamed *ZmHDA101*, *ZmHDA102*, and *ZmHDA108*) accumulate in the inner region of the endosperm with a nucleus-cytoplasmic localization during seed development [80]. Pull-down assays demonstrated that all these ZmRpd3 proteins can interact with ZmRBR1 and ZmRbAp1 proteins without competing in the binding. Similarities in the gene expression profiles and protein interactions indicated functional redundancy among members of the ZmRpd3 gene family, although a certain degree of functional divergence was also supported by the experimental evidence [80].

Recently, to investigate the functional redundancy between the *ZmHDA108* and *ZmHDA101* genes, *hda108* homozygous plants were crossed with plants of the AS33 antisense (AS) mutant line of the *hda101* gene [81]. The AS33 antisense line is characterized by a down-regulation of the *hda101* transcript and protein, which determines plant developmental defects and alteration of both the total level of histone acetylation and the transcription of the genes responsible for the vegetative-to-reproductive transition. The *hda108/+*; *AShda101* plants were selfed to obtain *hda108/hda108*; *AShda101* kernels. A strong defective kernel phenotype segregation was observed in the cobs. Approximately 25% of the selfing progeny presented a small collapsed nonviable kernel with a completely or partially empty pericarp [49], indicating that both genes participate in regulating endosperm-developmental processes during seed formation.

During endosperm development, chromatin and DNA modifications were analyzed in *Opaque2* targets [82]. The results of these analyses showed that transcriptional activation in the endosperm occurs through a two-step process, with an early potentiated state and a later activated state. The potentiated state is characterized by cytosine demethylation at symmetric sites, the substitution of H3K9me2 and H3K27me2 with histones H3 acetylated at Lys-14 (H3K14ac) and dimethylated at Lys-4 (H3K4me2), and increased DNaseI sensitivity. The active state showed a further increase of H3K14ac/H3K4me2 and DNaseI accessibility levels and the deposition of histone H3 acetylated at Lys-9 and trimethylated at Lys-4. These results indicated that mechanisms that modify chromatin states are involved in the *O2*-mediated regulation of transcription in the endosperm [82].

In plants, DNA methylation plays a role in the differential expression of genes depending on the sex of the parent that transmits them, a phenomenon called imprinting. The regulatory mechanism of imprinted gene expression implies that a cell can discriminate between genetically identical alleles differentially marked by an epigenetic imprint. In flowering plants, imprinting mainly occurs in the endosperm, with few cases of imprinted genes reported in the embryo (recently reviewed in [83]).

In maize, differentially methylated regions (DMRs) have been identified in the *α-tubulin* (*tubα3* and *tubα4*) and seed-storage protein genes (*α-zeins*), specifically in endosperm [54,55,84]. By analyzing different inbred lines and their reciprocal crosses, numerous conserved DMRs specific to the endosperm have been identified [85]. The DMRs were hypomethylated upon maternal transmission; conversely, upon paternal transmission, the endosperm methylation level was similar in the embryo and leaf. Maternal hypomethylation accounts for the 13% reduction in methyl-cytosine content of the endosperm compared with leaf tissue. DMRs were observed early after fertilization and maintained at a later stage of endosperm development [85].

The general picture emerging from all of this data is that seed development is achieved through the definition of specific compartments and endosperm cellular domains, and that this implies the presence of a network of genetic and epigenetic players coordinated in their activity. Genomically imprinted genes that are selectively expressed from the maternal genome are particularly attractive in maize endosperm where, unlike in both Arabidopsis and rice endosperm, little genome-wide demethylation occurs [86].

### 3.3. The Role of Auxin in Endosperm Development

The proper development of the three seed compartments, seed coat, endosperm, and embryo, depends on the coordination of the processes that lead to their differentiation, development, and maturation. The constant transmission and perception of signals by the three compartments coordinates all seed developmental processes. Phytohormones constitute one of these signals through the generation of their gradients and ratios in the different seed tissues [87]. Particularly relevant in this context are auxins, which maintain high levels of accumulation from fertilization to seed maturation (Figure 2).

In the endosperm, the aleurone, the basal endosperm transfer layer, and the embryo-surrounding region accumulate free auxin, which also has high levels in the kernel maternal chalaza [88]. The localization of the efflux auxin-carriers ZmPIN1 in maize endosperm indicated that both the transcript and protein-localization profiles overlap, and that *ZmPIN1* genes are expressed in the maize endosperm from the first developmental events (Figure 3). Whereas PIN1 proteins were shown to have a polar localization in the cell of different tissues of the maize plant [89], the ZmPIN1 proteins were never polarly localized in the plasma membrane of endosperm cells [88]. The role of auxin on endosperm development was further investigated in the maize *defective endosperm 18* mutant (*de18*), which has reduced levels of free and bound indole acetic acid (IAA) in the endosperm, leading to a reduction in dry matter accumulation [90]. Anti-IAA immunolocalization experiments disclosed a reduced accumulation of auxin in de18 endosperm and a lower expression of *ZmPIN1*, particularly so in the transfer cell layer [88]. These observations suggested that the downregulation of *ZmPIN1* expression and the lower auxin content in BETL cells might cause an alteration of transfer cell polarity and activity in nutrient-import into the developing endosperm, causing the reduced dry matter accumulation in *de18* kernels. Intriguingly, it was also observed that ZmPIN1 proteins, contrary to that reported in other cell types, are localized in the cytoplasm of the ESR cell domain, and are not targeted to the cell membrane. However, it is not known whether the ZmPIN1 accumulation of proteins in the internal cellular compartments is related to aspects of their function [88]. In the maize kernel, polar auxin transport always correlates with the differentiation of embryo tissues and the definition of the embryo organs [91]. The identification of nine additional *Zea mays* auxin efflux carriers *PIN* family members, and two maize *PIN-like* genes indicates that *ZmPIN2*, *ZmPIN5*, *ZmPIN8*, *ZmPIN10,* and one PIN-like gene named *ZmPINY* are expressed in maize endosperm, although their precise localizations and roles remain to be elucidated [92].

To further analyze the basis of IAA deficit in the *de18* mutant [90], the *ZmYuc1* gene was cloned and sequenced. *YUCCA* (*YUC*) genes encode flavin monooxygenases that catalyze the conversion of IPA to IAA [93]. Collective data indicated that *ZmYuc1* and *De18* are tightly associated, and that the aberrant YUC1 protein in *de18* is the causal basis of IAA deficiency and the small seed phenotype in that mutant [94].

Transcriptomic and metabolomic analyses of the maize *de18* mutant indicated that the IAA concentration controls sugar and protein metabolism during kernel differentiation, and that this is necessary for proper BETL differentiation [95]. To counteract the auxin deficit, the *de18* mutant adopts a fine-tuning of different auxin conjugates. At the transcriptional level, members of MYB and MADS-box gene families represent putative candidates as master regulators of the endosperm transcriptional regulation mediated by auxin. At the metabolic level, a link between auxin and storage protein accumulation was highlighted. This last observation suggests that IAA directly or indirectly, possibly through CK or ABA, might regulate the transcription of zein-coding genes [95].

All of these observations indicate that auxin regulation is important for maintaining the normal cell patterning of all endosperm domains, although it is reasonable to speculate that auxin crosstalk with other hormones could also influence maize endosperm tissue architecture. These research works conducted by Italian labs were fundamental for further international studies focused on the correct differentiation and patterning of maize kernel compartments, and strongly contributed to recent advances regarding hormones, sugars, receptors, and transcription factors involved in early maize development [31,96].

## 4. Conclusions

Maize is an important crop that provides both human food and animal feed. However, it is also a model species for basic and applied research. Both economic and scientific importance makes this species one of the first crops where cutting-edge technologies are usually applied. Additionally, the high level of genetic diversity of maize is challenging for all aspects of genetic studies. Currently, maize genomics provide breeders with exceptional tools for developing maize genotypes for a more sustainable agriculture and that are more adapted to a changing environment. Here, we have documented that for some decades Italian groups, mainly based in Northern Italy, have contributed to the improvement of the knowledge of maize genomics and to functional genetic studies through international collaborations. Particularly, during the last decades, insights into functional genomics have arisen from maize knockout/down mutants, which are fundamental in confirming the causal role between genotype and phenotype and deciphering the biological role of a gene. Of particular importance is the work carried out on seed formation and development through the characterization of mutants belonging to different international collections. Aside from their significance in basic research, these studies could be instrumental for the detection of key genes affecting grain filling process and reserve accumulation and thus influencing maize-yield, as well as nutritional quality [97,98,99]. In this context, they can have implications for breeding programs.

However, more recently in Italy, maize research has been limited by the scarcity of national and European fundings, although the curiosity and interest of Italian scientists in undertaking new studies on maize genetic complexity has remained unchanged.

## Figures and Tables

**Figure 1 ijms-23-11383-f001:**
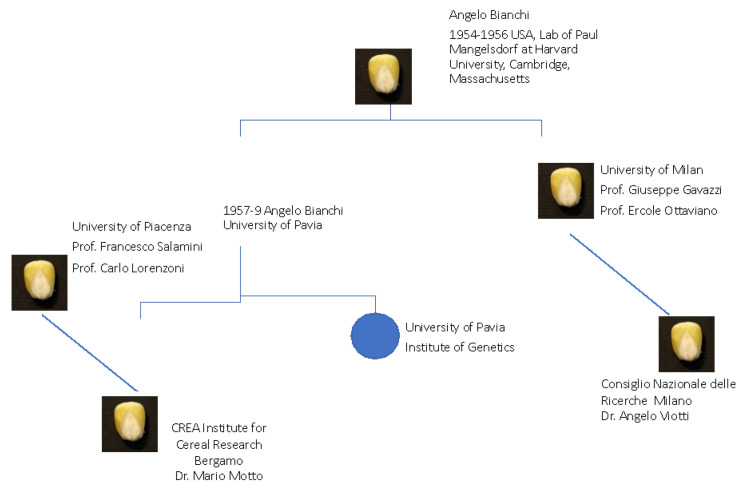
A summary picture of the Italian School Founders and their locations for maize research in Italy.

**Figure 2 ijms-23-11383-f002:**
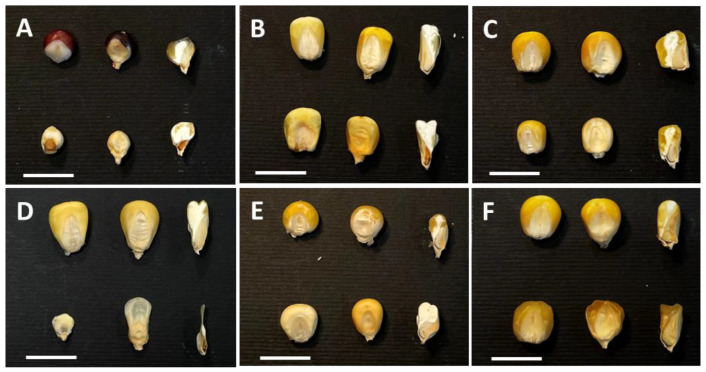
Representative images of seed mutant categories. In each panel, wild-type control (top) and mutant seeds (bottom) are shown. From left to right: pericarp-less, entire and longitudinally sectioned seeds. (**A**) *defective kernel*, *dek*; (**B**) *embryo-specific*; *emb*; (**C**) de; (**D**) *empty pericarp 4*, *emp4*; (**E**) *opaque*; and (**F**) *sugary* seed mutants. The bars correspond to 1 cm.

**Figure 3 ijms-23-11383-f003:**
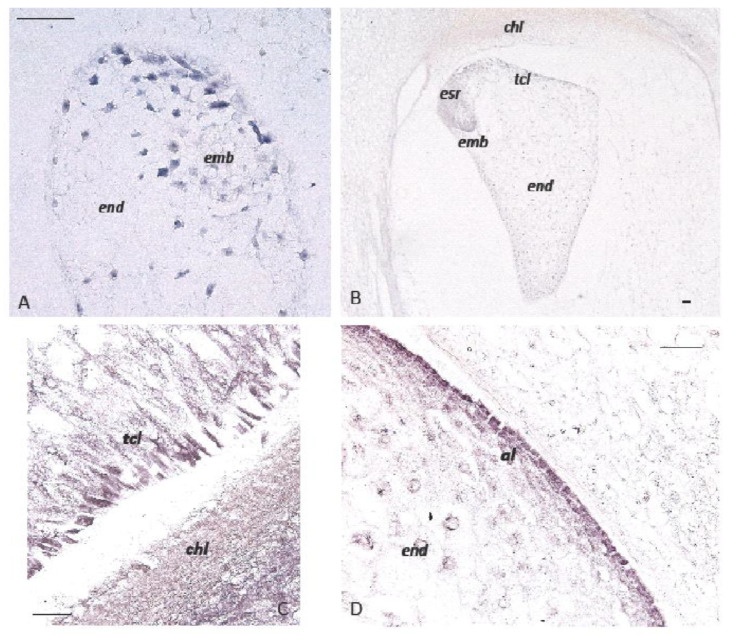
In situ hybridization of ZmPIN1 mRNAs and auxin accumulation during maize kernel development. All of the images represent longitudinal sections of B73 inbred kernels. Panels A and B show ZmPIN1 gene expression at an early stage of seed development; the blue color represents the hybridization signal. At 5 DAP, the ZmPIN1 gene starts to be expressed around the free dividing nuclei that form the syncytium and in small embryo (**A**). At the end of the cellularization phase around 10 DAP (**B**), the endosperm shows a gradient of ZmPIN1 expression. The transcripts are most abundant in the endosperm transfer cell layer (tcl), embryo-surrounding region (esr), and developing embryo (emb). Panels (**C**,**D**) show IAA-localization in the maize kernel at 16 DAP. An anti-IAA monoclonal antibody was employed to determine the auxin distribution and accumulation in developing maize endosperm. IAA accumulation (purple color) is detectable in the transfer cell layers (tcl) and maternal chalazal region facing the endosperm (end) in panel (**C**); in panel (**D**), auxin accumulation is also evident in endosperm (end) aleurone (al). Bars = 50 µm. (Forestan et al., 2010).

**Table 1 ijms-23-11383-t001:** The genes described in this review involved in seed and endosperm development in maize (*Zea mays*) and their orthologs in rice (*Oryza sativa*) are reported. Gene symbols and models were retrieved from the Maize Genetics and Genomics Database (MaizeGDB) (https://www.maizegdb.org/genome/genome_assembly/Zm-B73-REFERENCE-NAM-5.0, accessed on 26 August 2022) and the Rice Annotation Project Database (RAP-DB) (https://rapdb.dna.affrc.go.jp/index.html, accessed on 26 August 2022) for maize and rice, respectively.

*Zea mays.*				*Oryza sativa*	
Gene Name	Gene Symbol	Synonyms	Gene Model	Gene Symbol	Gene Model
*opaque endosperm2*	*op2*	*ZmbZIP1*	Zm00001eb301570	BZIP58	Os07G0182000
*floury2*	*fl2*	*zp19/22*-L34340*	Zm00001eb170070	none	none
*defective endosperm B30*	*de30*	*zp2*	Zm00001eb303160	none	none
*mucronate1*	*mc1*	*zp15**, *zp16*	Zm00001eb099950	none	none
*ribosome-inactivating protein1*	*b-32*	*rip1*	Zm00001eb350070	none	none
*Histone Deacetylase 101*	*Rpd3/101*	*rpd3*; *hda1*; *ZmHDA101*	Zm00001eb205290	HDAC3	Os02g0214900
*Histone Deacetylase 102*	*Rpd3/102*	*hda112*; *ZmHDA102*	Zm00001eb084120	none	Os04g0409600
*Histone Deacetylase 108*	*Rpd3/108*	*hda108*; *ZmHDA108*	Zm00001eb177560	none	Os08g0344100
*retinoblastoma-associated protein1*	*RbAp1*	*nfc102*; *rbap3*	Zm00001eb361300	WD40-23	Os01g0710000
*retinoblastoma-related protein*	*RBR1*	*rrb1*	Zm00001eb113470	none	Os11G0533500
*pyruvate orthophosphate dikinase-1*	*cyPPDK1*	*pdk1*	Zm00001eb287770	none	Os05G0405000
*defective endosperm 18*	*de18*	*yuc1*; *YUCCA1*	Zm00001eb409250	YUCCA11	Os12G0189500
*PIN-formed protein*	*PIN1*	*ZmPIN1a*	Zm00001eb372180	PIN1-like	Os06g0232300
*PIN-formed protein*	*PIN2*	*ZmPIN1b*	Zm00001eb254390	PIN1	Os02g0743400
*PIN-formed protein*	*PIN5*	*ZmPIN5a*	Zm00001eb143310	PIN5a	Os01G0919800
*PIN-formed protein*	*PIN8*	*none*	Zm00001eb154930	PIN8	Os01G0715600
*PIN-formed protein*	*PIN10*	*ZmPIN10a*	Zm00001eb158690	PIN10A	Os01G0643300
*PIN-formed like protein*	*PIN14*	*PINY*	Zm00001eb149720	none	Os01G0818000

## Data Availability

Not applicable.

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
