# Peer review of "The Italian Research on the Molecular Characterization of Maize Kernel Development"

_ijms, 2022, doi:10.3390/ijms231911383_

Round 1

Reviewer 1 Report

The main difficulty encountered when reviewing this manuscript is the lack of justification for an otherwise uncommon criterium for the restriction of its focus. The focus on the contribution of scientists from Italy, rather than focussing on hormonal, transcriptional or any other biological components regulating maize kernel development, is unexpected. The reader might fear that the ms might have overlooked significant contributions from non-Italian scientists. Nevertheless, the review is well written, clearly organized and informative, i.e. helpful. Only minor changes are suggested below:

Lns 143, 147, 217. The time window suggested for grain filling (15-45 DAP), and maturation (45-60 DAP) should be adjusted. Most scientists would agree that grain filling starts earlier (12 DAP) and, more importantly, finish earlier (30-35 DAP). The maturation phase is unlikely to extend to 60 DAP, since most researchers harvest the cobs before or at 50 DAP.

Suggestions to improve readability:

Ln60. Remove “have”

Ln62. Replace “have” with “has”

Ln68. Replace “holding” with “hosts”

Ln73. Replace “object” with “aim”

Ln77. Replace “after” with “Then”

Ln85. Replace “enables” with “facilitates”

Ln88. Delete “greatly”

Ln252. Replace “few” with “little”

Ln335. Replace “of the” with “up to”

Figure 2 legend. 3s substitute all Ts

Ln494. Replace “what” with “that”

Ln508. Delete sentence “In more detail, ZmYuc1 gene encoding the YUC1 protein is the causal basis of impairment in a critical step in IAA biosynthesis, essential for normal endosperm development in maize.” It looks redundant with the text immediately above.

Reviewer 2 Report

Review of MS "The Italian research on molecular characterization of maize kernel development" by Gabriella Consonni et al.  submitted to the IJMS

In this manuscript the authors assembled a very nice story and discussion of the contributions of Italian scientists to the field of maize caryopsis development. In their introduction they identify the foundational influence of Bianchi, Salamini, and Gavazzi to this very interesting field. They mention other important geneticists and molecular biologists that worked different aspects of maize research since the 1950s in Italy. This is only the prelude to a very well written description of advances in different areas of maize caryopsis development including a very detailed exposition of endosperm development, all focused in Italian contributions. I really enjoyed reading the manuscript and learnt that indeed there are different Italian "schools" that have trained students in the study of this very important scientific field that has a great importance for developmental biology in general and with consequential effects in agronomy and pant breeding. The text is very easy to understand thanks to the familiarity of the authors in the subject and to their great effort to communicate in a very straightforward manner the ins and outs of maize caryopsis development. In addition to very few minor corrections that the authors should take care of, I have a suggestion in terms of the historical background to the topic that would make even better the value of this fine review. Since this is a historical account of Italian research accomplishments in maize, it would be nice to make a brief summary of research interests in the 1950s that led to the foundation to the so called "Italian schools" in maize research. Angelo Bianchi seems to be a key player. Were there any other researchers before him that helped him and others to push their maize research careers?

I would like to see a reference to the book edited by Peterson and Bianchi in 1999 (Maize genetics and breeding in the 20th century. Editors, Peter A. Peterson, Angelo Bianchi. Singapore ; New Jersey : World Scientific 1999) where the contributors describe the distinct USA maize genetic "schools" and their relations to the Italian maize genetics community. A Figure with the main nodes (school founders), their students and their interactions with other "schools", American or European, could help us to quickly understand their influence in Italian and in International science. It seems that Felice Ippolito played a very influential role during the birth  of the Milan school, as Secretary of CNFN (National Committee for Nuclear Energy, today´s ENHA) as he made possible the hiring of Angelo Bianchi at Milan University’s Institute of Genetics, according to Sari-Gorla (MAYDICA 1994, Vol. 39, No. 2). How Ippolito´s decisions helped shape the path of the Italian research community after the 1950s?

Minor points for correction:

- In line 33: "....maize (Zea mays L.)..". Scientific names should be written in italics.

- Line 60: "......in phytic-acid biosynthesis and have ....". Delete "and"  

- Page 6. Figure 1 Legend. Define the mutant names as in the main text: dek (defective kernel), etc.

- Lines 269-270: - Are there any web sites that direct the reader to these repositories: "...mutagenesis-induced mutants, is still maintained and exploited at Università di Piacenza. Mutant collections are also present at Università degli Studi di Milano and Università di Bologna."

- Line 302: "...to the beginning of 80s, " Change to: "...to the beginning of the 80s, "

- Line 335: ".. zein proteins are reduced of the 60% and the 22kd zein class component is almost absent". The authors mean that zein proteins are reduced to 60% of the wt levels or reduced by 60% (40% of the wt levels)?

- Lines 454  and 455: The following sentence sojnd awkward and confusing: "...Genomically imprinted genes that are selectively expressed from the maternal genome in endosperm are particularly attractive in maize endosperm,"
